# Magnetostructural Properties of Some Doubly-Bridged Phenoxido Copper(II) Complexes

**DOI:** 10.3390/molecules28062648

**Published:** 2023-03-14

**Authors:** Salah S. Massoud, Febee R. Louka, Madison T. Dial, Nahed N. M. H. Salem, Roland C. Fischer, Ana Torvisco, Franz A. Mautner, Kai Nakashima, Makoto Handa, Masahiro Mikuriya

**Affiliations:** 1Department of Chemistry, University of Louisiana at Lafayette, P.O. Box 43700, Lafayette, LA 70504, USA; 2Department of Chemistry, Faculty of Science, Alexandria University, Moharam Bey, Alexandria 21511, Egypt; 3Institut für Anorganische Chemische, Technische Universität Graz, Stremayrgasse 9/V, A-8010 Graz, Austria; 4Institut für Physikalische and Theoretische Chemie, Technische Universität Graz, Stremayrgasse 9/II, A-8010 Graz, Austria; 5Department of Chemistry, Interdisciplinary Graduate School of Science and Engineering, Shimane University, 1060 Nishikawatsu, Matsue 690-8504, Japan; 6Department of Applied Chemistry for Environment, School of Biological and Environmental Sciences, Kwansei Gakuin University, 1 Gakuen Uegahara, Sanda 669-1330, Japan

**Keywords:** copper(II) complexes, phenolate compounds, tripodal ligands, X-ray structures, magnetic properties, computational

## Abstract

Three new tripod tetradentate phenolate-amines (**H_2_L^1^**, **H_2_L^4^** and **H_2_L^9^**), together with seven more already related published ligands, were synthesized, and characterized. With these ligands, two new dinuclear doubly-bridged-phenoxido copper(II) complexes (**3**, **4**), and six more complexes (**1**, **2**, **5**–**8**), a new trinuclear complex (**9**) with an alternative doubly-bridged-phenoxido and –methoxido, as well as the 1D polymer (**10**) were synthesized, and their molecular structures were characterized by spectroscopic methods and X-ray single crystal crystallography. The Cu(II) centers in these complexes exhibit distorted square-pyramidal arrangement in **1**–**4**, mixed square pyramidal and square planar in **5**, **6**, and **9**, and distorted octahedral (5+1) arrangements in **7** and **8**. The temperature dependence magnetic susceptibility study over the temperature range 2–300 K revealed moderate–relatively strong antiferromagnetic coupling (AF) (|*J*| = 289–145 cm^−1^) in complexes **1**–**6**, weak-moderate AF (|*J*| = 59 cm^−1^) in the trinuclear complex **9**, but weak AF interactions (|*J*| = 3.6 & 4.6 cm^−1^) were obtained in **7** and **8**. No correlation was found between the exchange coupling *J* and the geometrical structural parameters of the four-membered Cu_2_O_2_ rings.

## 1. Introduction

Dinuclear copper(II) complexes in which the Cu(II) ions are bridged by a phenolate oxygen atoms (L-O) and an exogenous bridging atom X with the coordination core Cu(µ-L-O)(µ-X) Cu (X = L-O, OH^−^, OR^−^, O^2−^, OAc^−^, N_3_^−^and Cl^−^) have been reported for a number of ligands with different skeletons frames [1,2,3,4,5,6,7,8,9,10,11,12]. Some of these complexes have served as cooperative model systems to mimic the active sites in metalloproteins, such as hemocyanin [12,13,14,15,16], metalloenzymes, such as tyrosinase [16,17,18], galactose oxidase (GOase) [19,20,21,22,23,24], superoxide dismutase (SOD) [25,26,27], and as artificial nucleases for promoting the cleavage of DNA [5,6,28,29] as well as anticancer agents [30,31].

The dinuclear copper(II) compounds constructed from the cores Cu(µ-L-O)(µ-X)Cu and Cu(µ-L-O)_2_Cu are stable and the two metal ions are located in close proximity, which lead to strong implications regarding their reactivity and magnetic properties due to the interactions between the two paramagnetic Cu(II) centers [32,33]. Different strategies have been employed in the design of this class of compounds. The first approach was achieved though the interaction Cu(II) salts with tridentate ligand-based phenolates in their skeletons [34,35,36,37,38,39,40,41], and the second approach was via the design of bicompartmental phenolates bearing two pendent arms of N-donor groups [1,2,3,4,5,6,42,43]. A third approach was also used through the design of tripodal pyridyl tetradentate ligands containing one or two phenolate arms [12,14,15,20,29,44,45,46,47,48,49,50,51].

The structures and magnetic properties of singly-bridged Cu(µ-L-O)Cu and doubly-bridged phenolate Cu(µ-L-O)_2_Cu compounds were the subject of numerous investigations [1,2,3,4,31,33,34,41,42,44] as well as theoretical studies [31,33,41,51,52,53,54]. The energy gap 2*J* arising from the spin singlet-triplet (S-T) state interactions between the two local Cu(II) doublets were evaluated. Many structural parameters were reported to affect the magnitude of the magnetic interactions in the four membered ring, Cu_2_O_2_ [41,54]. These include the Cu···Cu bond distance and the bridged Cu−O−Cu bond angle [41,54], the geometrical distortion, the dihedral angle between the two copper planes as well as the electronegativity in the bridging phenolate [53]. The presence of electron-withdrawing groups leads to substantial reduction in the antiferromagnetic exchange [53]. The steric effect incorporated into ligand skeleton, which may distort the copper coordination geometry, cannot be ruled out either. The non-covalent bonding interactions (H-bonding, π···π, stacking interactions, and van der Waals’ force) have also been reported to affect the molecular magnetism in these systems [54,55,56,57,58,59,60,61,62]. In general, strong super exchange antiferromagnetic coupling (AF) was observed in the bridged phenolate compounds [25,26,27,33,41,43,51,52,53,54] but, in a few cases very weak ferromagnetic properties (F) were found [4,22,25,54]. 

Herein, we describe a general effective procedure for the design of novel series of tripodal phenolate compounds containing pyridyl or aliphatic amine arms with N_3_O and N_2_O_2_ chromophores, where the donor O atoms are provided by one or two phenolate groups (Figure 1) and their corresponding doubly-bridged phenoxido-Cu(II) complexes. The magnetic properties of the complexes were investigated at variable temperatures and the evaluated exchange coupling constants, *J*, are discussed in relation to their molecular structures. 

## 2. Results and Discussion 

### 2.1. Synthesis of the Ligands and Complexes

The new tetradentate tripodal-phenolate amines (**H_2_L^1^**, **H_2_L^4^** and **H_2_L^9^**) used in this work, together with related ligands (Figure 1), were synthesized by a standard general procedure. In a typical experiment, a methanolic mixture containing 2,4-disubstituted phenol and the corresponding amine (2:1), as well as two molar amounts of an aqueous 37% HCHO and triethylamine (Et_3_N) are used, whereas in the case of 2-methoxy-phenols and pyridyl derivatives (**HL^7^** and **HL^8^** ligands, Figure 1), equimolar ratios were employed for all reagents. Refluxing the solutions for 3 to 4 days, followed by evaporation of the solution resulted in the formation of the solid products in reasonable yields, but about 30% were observed in the bromo-phenolate **H_2_L^3^**. We should mention that too much evaporation of the resulting solutions may lead to the formation of an oily product and/or a semi-solid, which is hard to recrystallize and leads to impure products. The pure desired products were obtained upon recrystallization from ethyl acetate and activated charcoal (see Experimental section). The synthesis and characterization of **H_2_L^2^**, **H_2_L^3^**, **H_2_L^5^**, **H_2_L^6^, HL^7^, HL^8^,** and **H_2_L^10^** were recently reported by our group [30].

The doubly-bridged phenoxido Cu(II) complexes **1**–**6**, **9**, and the polymeric 1D **10** were obtatined by the reaction of a methanolic solution of Cu(NO_3_)_2_·3H_2_O with the corresponding tripodal-phenolate amine ligand and Et_3_N in the stochiometric 1:1:2 molar ratio. In the cationic complexes **7** and **8**, copper(II) perchlorate was used with the bipyridyl-phenolates **HL^7^** and **HL^8^**, and Et_3_N (1:1:1), respectively. These reactions afforded the desired complexes in moderate to good yields (55–90%). An illustration for the synthesis of the doubly-bridged phenoxido moieties in the complexes derived from N,N-dialkylethylenediamines is shown in Figure 2. Interestingly, although all double bridged phenoxido compounds display green or olive-green color, brownish-green and purple colored complexes were obtained in case of the trimeric complex **9** and the 1D-polymeric **10**. Single crystals suitable for X-ray structural determination were obtained either from dilute methanolic solution and/or recrystallization from CH_3_CN or acetone. The complexes are slightly soluble in MeOH, but are more soluble in less polar solvents, such as CH_3_CN and DMSO. The isolated compounds were characterized by elemental microanalyses, spectroscopic techniques, and conductivity measurements as well as by single X-ray crystallography for the copper complexes.

### 2.2. Characterization of the Ligands

Some very general features exist in the IR spectra of the tetradentate tripodal phenolate amine ligands, such as a very weak broad band or a shoulder over the frequency region 3140–3230 cm^−1^, which was attributed to the stretching vibration, ν(O-H) of the phenolic groups, in addition to a series of weak to very weak bands observed over the range 2700–3050 cm^−1^ due to ν(C-H) stretching of the aliphatic and aromatic groups. The moderate intense band detected around the 1590–1600 cm^−1^ region is assigned to ν(C=N), whereas the moderate to strong series of bands shown in 1200–1590 cm^−1^ are most likely attributed to ν(C=C, C-O). The ESI-MS of the ligands in MeOH, showed the 100% *m/z* base peak that corresponds to the protonation of parent ligand (H_2_L^n^ + H]^+^ (n =2–5, 9, 10). The ^1^H NMR (*d_6_*-DMSO) spectra displayed peak positions at 7.1–6.8 (protons-ph); 5.0–4.9 (phenolate-protons), 3.8–3.7 (CH_2_-py); 3.5–3.3 (CH_2_-ph); 2.5–2.1 ppm (CH_3_-ph). The pyridyl protons reveal their signals at δ = 8.7–7.1 in compounds **H_2_L^1^**, **HL^4^** and **HL^8^**. The hydroxyl phenolate protons were clearly observed in **H_2_L^1^** and **H_2_L^5^** compounds but was not seen in most compounds, most likely due to their low solubility in DMSO. 

### 2.3. Characterization of the Complexes

The IR spectra of the complexes under investigation were almost similar to the spectral pattern observed in their parent ligands, except for the disappearance of the broad band or shoulder in the region 3140–3230 cm^−1^ of the ν(O-H) of the phenolic groups upon its deprotonation and/or coordination to the Cu^2+^ ion. The two cationic perchlorate complexes **7** and **8** displayed a very strong band at 1079 and 1076 cm^−1^, respectively, due to ν_as_(O-Cl) of the perchlorate counter ions. The O-H stretching frequency, ν(O-H), for the water of crystallization in **5** and **6** and were shown as a broad band over the 3320–3400 cm^−1^ region. The ESI-MS (CH_3_CN) of the cationic pyridyl complexes **7** and **8** revealed the monomeric species with *m*/*z* = [Cu(**L^7,8^**)]^+^ (100%), where the base peak (100%) of the remaining complexes was consistent with the release of the parent protonated ligands with *m*/*z* = [**H_2_L^2−6,9,10^** + H]^+^. 

The UV-vis spectra of the complexes under investigation, measured in DMSO and/or CH_3_CN at room temperature, displayed a single broad/shoulder band in the 610–750 nm region and a strong absorption band over wavelength region 410–500 nm. The latter intense band can be assigned to the bridged phenoxido charge transfer transition (L-O → Cu^II^ LMCT) [12,20,63,64], whereas the former low intense broad band is attributed to d-d transition in five-coordinate Cu(II) complexes, which was occasionally accompanied with a weaker intense broad band around 850–890 nm. The d-d transition feature in solution is consistent with a distorted square pyramidal geometry (SP) around the central Cu(II) ion [32,48,49]. Thus, the distorted SP geometrical assignments observed in DMSO, CH_3_CN, or acetone solution, were retained in the solid state (see X-ray section). The solution spectra of the complexes in these media did not show any appreciable change over the one-week period, reflecting their high stability. 

The position of λ_max_ in the 610–750 nm region can be used as a criterion for the ligand field strength of the tripodal phenolate ligands; λ_max_ decreases in the order: **9** (λ_max_ = 748 nm) > **2** (λ_max_ = 733 nm) > **5** (λ_max_ = 708 nm) > **10** (λ_max_ = 687 nm) > **6** (λ_max_ = 655 nm) > **4** ≈ **8** (λ_max_ = 630 nm) > **7** (λ_max_ = 625 nm) > **1** (λ_max_ = 620 nm). This means that the ligand field strength is decreasing in the reverse order and the strongest fields are for phenolate compounds containing pyridyl arms. However, we should mention that broadening and close location of this band in some complexes did not permit precise prediction of their actual ligand field strength, but results are consistent with previous results [65,66]. In general, ligand field strengths decrease with increasing chelate ring size, the presence of electronegative chlorine or bromine atoms in the phenolate groups, and/or steric effect on the coordinated N-aliphatic amine arms, which tends to reduce the electron density on the coordinated centers and hence its ligand field [65,66]. 

The non-electronic nature of all complexes, with the exception of **7** and **8**, was supported by measuring the molar conductivities, Λ_M_ in CH_3_CN or DMSO, whenever the solubility permits. The measured Λ_M_ values were ≤ 5 Ω^−1^·cm^2^·mol^−1^, which were fully consistent with their non-ionic properties as predicted by their molecular formulas. The measured Λ_M_ values (DMSO) of the perchlorate complexes **7** and **8** were within the range of 280 ± 2 Ω^−1^·cm^2^·mol^−1^. These values are in full agreement with the 1:2 electrolytic behavior of the doubly-bridged phenoxido compounds ([Cu_2_(µ_2_-**L^7,8^**)_2_](ClO_4_)_2_ (**7**, **8**) [30,67]. 

### 2.4. Description of the Structures of Complexes 

The crystal structure of title compounds **3** and **4** consist of neutral dinuclear [Cu_2_(**L^3^**)_2_] and [Cu_2_(**L^4^**)_2_] units, respectively; the latter co-crystallizes with an acetone solvent molecule per dinuclear unit (Figure 1,b). In both structures, the phenolate oxygen atoms O2 and O4 of two tripod ligand **L^3^** or **L^4^** anions are bridging the two copper metal centers to form four-membered Cu_2_O_2_ rings [Cu-O from 1.9465(16) to 2.0349(16) Å, Cu1 Cu2 = 3.047(2) (**3**) and 2.9949(4) Å (**4**); Cu1-O-Cu2 from 97.17(7) to 100.4(4)°, O-Cu-O from 74.8(4) to 75.78(7)°] (Table 1). Each Cu1 center is penta-coordinated with further three terminal sites occupied by the amine N1 and N2 atoms as well as the phenolate O1 atom. Each Cu2 center is also penta-coordinated with amine N3, N4, and phenolate O3 atoms. While the CuN_2_O_3_ chromophore of Cu1 may be described as less distorted square pyramid (SP) with a τ_5_ -value of 0.06; the CuN_2_O_3_ chromophore of Cu2 is strongly distorted SP with τ_5_ -value of 0.21 (**3**) and 0.45 (**4**) (τ = 0 for ideal SP) and τ_5_ = 1 for ideal trigonal bipyramid (TBP)) [68]. Their apical positions are occupied by N2 and N4 atoms [Cu-N(apical) from 2.180(10) to 2.449(2) Å]. The Cu-N1/N3(basal) bonds vary from 2.0627(19) to 2.160(10) Å, and the Cu-O1/O3(basal) bonds from 1.8783(17) to 1.905(8) Å. The trans-basal bond angles vary from 138.62(7) to 165.55(7)°. 

The non-planar Cu_2_O_2_ four-membered rings have hinge distortion expressed by their Cu-O-Cu-O dihedral angles of 24.3° and 26.9° for **3** and **4**, respectively. The phenoxido groups deviate from phenyl out-of-plane angles (τ) of 10.5 and 12.2° in **3**, and of 3.2 and 12.8° in **4**, and phenyl ring torsion angles Cu-O-C-C of 50.9 and 71.0° in **3** and 52.5 and 67.5° in **4**.

The crystal structure of **9** features trinuclear complex units (Figure 1c) and partially disordered MeOH solvent molecules. The phenolato O2 and O5 atoms of two tripod ligand L^9^ anions and O3 and O4 of two MeO^−^ anions are bridging the central Cu2 center with the two external Cu1 and Cu3 centers to form two four-membered Cu_2_O_2_ rings [Cu-O from 1.920(2) to 2.000(2) Å, Cu1···Cu2 = 2.9652(5), Cu2···Cu3 = 2.9418(5) Å; Cu1···Cu2···Cu3 = 154.63(2)°; Cu-O-Cu from 96.48(9) to 100.19(9)°, O-Cu-O from 75.43(8) to 77.91(8)°] (Table 1). The four oxygen atoms around Cu2 center form a tetragonally distorted square planar CuO_4_ geometry with a τ_4_-value of 0.17 (τ_4_ = 0 for ideal square planar (SQP) and τ_4_ = 1 for ideal tetrahedral, (*T_d_*) geometry) [9]. Coordination number 5 around the Cu1 and Cu3 center is completed by two N and one O donor atoms of two tripod **L^9^** anions. Their CuN_2_O_3_ chromophores form SP geometry with τ_5_(Cu1) of 0.01 and τ_5_(Cu3) of 0.23. Their apical positions are occupied by N2 and N4 atoms [Cu1-N2 = 2.351(2), Cu3-N4 = 2.324(2) Å]. The Cu-N1/N3(basal) bonds are 2.054(2) and 2.066(2) Å, and the Cu-O1/O3(basal) bonds are 1.929(2) and 1.944(2) Å. The trans-basal bond angles vary from 153.32(9) to 167.39(9)°. The non-planar Cu_2_O_2_ four-membered rings have hinge distortion expressed by their Cu-O-Cu-O dihedral angles of 22.9° and 23.5°. The phenoxido groups deviate from phenyl out-of-plane angles, τ of 8.2 and 20.0°, and phenyl ring torsion angles Cu-O-C-C of 44.2 and 41.9°. The trinuclear subunit may formally be created by the insertion of a “Cu(MeO)_2_” moiety in the center of a bis(phenolato)-bridged Cu_2_O_2_ four-membered ring of a dinuclear compound (e.g., of **3**).

Perspective views of the dinuclear complexes **1**, **2**, **5–8** containing the Cu_2_O_2_ bis(phenolato)-bridged subunits, as well as a section of the polymeric chain of **10**, are presented in Figure 2. The crystal structures have been reported previously, along with their anticancer properties [30]. 

### 2.5. Magnetic Properties 

According to the crystal structures, the complexes under investigations can be divided into six groups as follows: 

Group 1: Centrosymmetric di-µ-phenoxido-bridged dinuclear complexes with a distorted square-pyramidal geometries (**1** and **2**). Complex [Cu_2_(**L^1^**)_2_] (**1**) shows a distorted square-pyramidal arrangement (SP) (*τ_5_* = 0.23) around each copper(II) center with an axial Cu-N distance of 2.310(3) Å, Cu···Cu distance and Cu-O-Cu angle are 3.0915(7) Å and 103.44(9)°, respectively [30]. Thus, Complex **1** is expected to exhibit antiferromagnetic interaction (AF) between the two *S* = ½ spins through the two phenoxido bridges. Temperature dependence of magnetic susceptibilities, measured under an external magnetic field of 0.5 T, ranging from 2 K to 300 K are represented in Figure 3 in the form of *χ*_M_ and *µ*_M_ *vs T* plots, where *χ*_M_ is the magnetic susceptibility per Cu_2_ unit, *µ*_M_ is the magnetic moment per Cu_2_ unit, and *T* is the absolute temperature. The magnetic moment of 1.30 BM at 300 K is considerably lower than the spin-only value (2.45 BM) for two non-interacting copper(II) *S* = ½ ions. The magnetic moment exhibits a continuous decrease with lowering the temperature over the range 2–300 K and reaches 0.28 BM at 2 K. This magnetic behavior reveals significant AF interaction between the copper(II) ions. The magnetic data were analyzed by the Bleaney–Bowers equation based on the Heisenberg model: *χ*_M_ = (1 − *p*)(2*Ng*^2^*µ*_B_^2^*/kT*)[3 + exp(−2*J*/*kT*)]^−1^ + *pNg*^2^*µ*_B_^2^/2*kT* + 2*Nα*(1)
where *g* is the *g* value, *J* is the exchange coupling constant between the two copper(II) ions, *p* is the fraction of mononuclear copper(II) impurity, and *Nα* is the temperature-independent paramagnetism for each copper(II) ion [8]. The best-fitting parameters (*g* = 2.1 (fixed), 2*J* = −578 cm^−1^, *p* = 0.016, and *Nα* = 60 × 10^−6^ cm^3^ mol^−1^ (fixed)) are in complete support for strong AF interaction.

The crystal structure of [Cu_2_(**L^2^**)_2_] (**2**) also shows the same crystallographic similarity as **1** (*τ* = 0.24) [68] around each copper(II) center with the axial Cu-N distance of 2.3580(17) Å. The Cu···Cu distance and Cu-O-Cu angle are 3.0064(5) Å and 99.12(6)°, respectively. The *µ*_M_ of **2** is 1.62 BM per dinuclear unit at 300 K, which is also considerably lower than the spin-only value and decreases with lowering of the temperature and reaching to 0.12 BM at 2 K. The best-fitting parameters (*g* = 2.10, 2*J* = −403 cm^−1^, *p* = 0.0007, and *Nα* = 60 × 10^−6^ cm^3^ mol^−1^) disclose the expected strong AF interaction. The absolute 2*J* value is smaller than that of **1**, but it reflects the strong AF interaction in **1** compared **2** and is in harmony with the associated wider Cu-O-Cu angle observed in **1.**

Group 2: Pseudo-symmetric di-µ-phenoxido-bridged dinuclear complexes with distorted SP geometries (**3** and **4**). The crystal structure of [Cu_2_(**L^3^**)_2_] (**3**) showed that the complex exhibits distorted SP arrangements (*τ* = 0.06 and 0.37) around two copper(II) centers with axial Cu-N distances of 2.180(10) and 2.235(10) Å. The Cu···Cu distance is 3.047(2) Å and Cu-O-Cu angles are 98.8(4) and 100.4(4)°. Complex **3** crystallizes in non-centrosymmetric with different distortion around each copper(II) center. The magnetic properties of **3** is illustrated in Figure 4. The magnetic moment of 2.07 BM at 300 K is a little higher than those of **1** and **2**, and it exhibits a gradual decrease with lowering the temperature, reaching 0.20 BM at 2 K. The magnetic data were analyzed by Equation (1), and the obtained best-fitting parameters (*g* = 2.14(1), 2*J* = −291 cm^−1^, *p* = 0.009, and *Nα* = 60 × 10^−6^ cm^3^ mol^−1^ (fixed)) demonstrates the AF interaction between the two copper(II) centers. The absolute 2*J* value of **3** is smaller than those of **1** and **2**, but with a weaker interaction. This may be attributed to the different distortion between the two copper(II) coordination environments, which causes poor overlap between the two magnetic orbitals compared to **1** and **2**. 

The complex [Cu_2_(µ_2_-**L^2^**)_2_] (**4**) exhibits crystallographic similarities comparable to **3**. The molecule is a non-centrosymmetric di-µ-phenoxido-bridged complex with distorted SP arrangements (*τ_5_* = 0.07 and 0.45) around the central copper(II) centers with the axial Cu-N distances of 2.449(2) and 2.3532(19) Å. The Cu···Cu distance and Cu-O-Cu angles are 2.9949(4) Å and 97.17(7) and 97.78(7)°, respectively. The magnetic moment *µ*_M_ (1.98 BM at 300 K) shows a gradual decrease with lowering the temperature, reaching 0.20 BM at 2 K. The magnetic data were analyzed by Equation (1) and the obtained best-fitting parameters (*g* = 2.10 (fixed), 2*J* = −293 cm^−1^, *p* = 0.008, and *Nα* = 60 × 10^−6^ cm^3^ mol^−1^ (fixed)) showed a comparable AF interaction to that observed in **3**. This is in accordance with the relationship between the 2*J* and Cu-O-Cu values. 

Group 3: Unsymmetric di-µ-phenoxido-bridged dinuclear complexes with distorted square-pyramidal (SP) and square-planar (SQP) geometries (**5** and **6**). The crystal structure of complex [Cu_2_(L^5^)_2_(H_2_O)]·2H_2_O (**5**) showed that the molecule exhibits a distorted SP copper(II) (*τ_5_* = 0.011) with an axial Cu-O bond of 2.538(3) Å, and a distorted SQP copper(II) with the *τ*_4_ value of 0.17. Thus, if we consider the long axial distance of 2.538(3) Å, then both of the magnetic orbitals of the two copper(II) centers are expected to exist within the basal NO_3_ and NO_3_ square plane, favoring a strong magnetic interaction. The Cu···Cu distance and Cu-O-Cu angles are 2.9331 Å, and 96.70(9) and 97.58(9)°, respectively. The magnetic moment, *µ*_M_ (1.82 BM at 300 K), is lower than the spin-only value (2.45 BM) for two non-interacting copper(II) *S* = ½ ions. The magnetic moment revealed a gradual decrease with lowering the temperature and reaches 0.31 BM at 2 K (Figure 5). The best-fitting parameters to Equation (1) (*g* = 2.1 (fixed), 2*J* = −352 cm^−1^, *p* = 0.036, and *Nα* = 60 × 10^−6^ cm^3^ mol^−1^ (fixed)) are consistent with considerable AF interaction between the two copper(II) ions. The absolute 2*J* value is comparable to those obtained in **3** and **4**. 

The crystal structure of [Cu_2_(**L^6^**)_2_(H_2_O)]·2H_2_O (**6**) shows crystallographic parameters similar to **5**, having a distorted SP copper(II) (*τ_5_* = 0.013) with the axial Cu-O bond of 2.512(2) Å and a distorted SP copper(II) with *τ_4_* value of 0.17. The Cu···Cu distance and Cu-O-Cu angles are 2.9410(4) Å and 97.56(7) and 96.93(7)°, respectively. The magnitude of the magnetic moment of **6** is 1.68 BM at 300 K. This value is lower than the spin-only value for copper(II), *S* = ½ ion, and the magnetic moment exhibits a continuous decrease with lowering temperature and reaches 0.11 BM at 2 K. The best-fitting parameters to Equation (1) (*g* = 2.10 (fixed), 2*J* = −500 cm^−1^, *p* = 0.0009, and *Na* = 170 × 10^−6^ cm^3^ mol^−1^) showed a significant AF interaction between the two copper(II) ions. The magnetic behavior of **6** should be considered as similar to that of **5**, irrespective of the relatively large –*J* value of **5**, taking into consideration the similarity of the Cu-O-Cu angle, Cu-O-Cu-O dihedral angle, phenyl ring torsion angle (Cu-O-C-C angle), and phenyl out-of-plane shift angle t (Table 2), then one should expect strong AF interaction between the copper(II) centers in the two compounds **5** and **6**. 

Group 4: Centrosymmetric di(phenoxido)-bridged dinuclear complexes with a distorted (5 + 1) octahedral geometries (**7** and **8**). The crystal structural analysis of [Cu_2_(**L^7^**)_2_](ClO_4_)_2_ (**7**) shows a distorted octahedral (5 + 1) geometry with a semi-coordination Cu-OMe of 2.785(4) Å around each copper(II) center. If this semi-coordination is neglected as a substantial bond because of its long distance, then the coordination geometry can be regarded as a distorted square-pyramidal geometry with the τ_5_ value of 0.41. Therefore, taking into consideration the long distance of the axial Cu-O(phenoxido) bond of 2.213(3) Å, as a result the magnetic orbital of each copper(II) ion may exist around the basal N_3_O plane mainly and not in the direction of the axial phenoxido-bridging bond, which leads to a poor magnetic interaction. The Cu···Cu distance and Cu-O-Cu angle are 3.130 Å and 97.72 (14)°, respectively. The magnetic data of **7** are illustrated in Figure 6. The µ_B_ (2.65 BM at 300 K) is higher than the spin-only value (2.45 BM) for two non-interacting copper(II), *S* = ½ ions. The magnetic moment exhibits a continuous decrease with lowering temperature and reaches 0.51 BM at 2 K, suggesting an AF interaction. The best-fitting parameters (*g* = 2.14, 2*J* = −7.2 cm^−1^, and *Nα* = 60 × 10^−6^ cm^3^ mol^−1^ (fixed)) are consistent with a weak AF interaction between the two copper(II) ions.

Complex [Cu_2_(**L^8^**)_2_](ClO_4_)_2_ (**8**) exhibits very close crystallographic parameters like **7**; distorted octahedral (5 + 1) arrangement with a semi-coordination of Cu-OMe of 2.791(3) Å around each copper(II) ion. The coordination geometry of this complex is similar to **7** and can be regarded as indicated above as a distorted SP geometry (τ_5_ value of 0.40) with a relatively long axial phenoxido-bridging of Cu-O distance of 2.215(2) Å. The Cu···Cu distance and the Cu-O-Cu angle are 3.098(2) Å and 96.63(10)°, respectively. The similarity was also extended to the magnetic properties of **7**. The best-fitting parameters (*g* = 2.1 (fixed), 2*J* = −9.2 cm^−1^, and *Nα* = 60 × 10^−6^ cm^3^ mol^−1^ (fixed)) shows a weak AF interaction between the two copper(II) ions.

In **7** and **8**, the Cu_2_O_2_ bridging ring with the axial phenoxido-bridges is planar, as can be seen from the Cu-O-Cu-O dihedral angle of 0° (Table 2), and thus the magnetic orbitals of each copper(II) ion should be parallel to each other. This suggests that the phenoxido oxygens are rather axially located relative to the basal dx^2^-y^2^ orbital plane. Such axial bridging oxygens cannot effectively participate in mediating the magnetic interaction through the phenoxido-bridges.

Group 5: linear trinuclear complex **9**. The crystal structure of [Cu_3_(**L^9^**)_2_(µ-OCH_3_)_2_]·CH_3_COCH_3_ (**9**) shows that the molecule has a µ-phenoxido-µ-methoxido-bridged trinuclear copper(II) complex with a linear arrangement of the three copper(II) ions. The magnetic data (Figure 7) revealed a magnetic moment of 2.50 BM at 300 K, which is lower than the spin-only value (3.00 BM) for three non-interacting copper(II), *S* = ½ ions. The magnetic moment exhibits a gradual decrease with lowering temperature and reaches 1.46 BM at 2 K. This behavior suggests a remarkable AF interaction between the copper(II) ions. The magnetic analysis was carried out with the susceptibility equation for the linear trinuclear copper(II) ions based on H = −2*J*_Cu3_(*S*_1_·*S*_2_ + *S*_2_·*S*_3_). However, the fitting was not good, and rather better fitting was obtained when the magnetic data were analyzed by the dinuclear model using Equation (1). The results indicated that the magnetic data of **9** contain some impurities arising from the presence of dinuclear copper(II) species. Therefore, the magnetic data were analyzed with the aid of Equation (2) of linear trinuclear and dinuclear copper(II) model.
*χ*_M_ = (1 − *p*_Cu2_)(*Ng*^2^*µ*_B_^2^*/*4*kT*)[1 *+* exp(−2*J*_Cu3_/*kT*) + 10exp(*J*_Cu3_/*kT*)]/[1 + exp(−2*J*_Cu3_/*kT*) + 2exp(*J*_Cu3_/*kT*)] + *p*_Cu2_(2*Ng*^2^*µ*_B_^2^*/kT*)[3 *+* exp(−2*J*/*kT*)]^−1^ + (3−*p*_Cu2_)*Nα*(2)
where *p*_Cu2_ is the fraction of dinuclear copper(II) impurity. The best-fitting parameters (*g* = 2.1 (fixed), *J*_Cu3_ = −59 cm^−1^, *p*_Cu2_ = 0.315, 2*J* = −592 cm^−1^, and *Nα* = 60 × 10^−6^ cm^3^ mol^−1^ (fixed)) revealed the existence of a considerable amount of dinuclear copper(II) species in the sample.

Group 6: polynuclear chain complex **10**. The X-ray structure analysis revealed that catena-[Cu(µ-**L^10^**)] (**10**) is a polynuclear chain molecule constructed of tetrahedrally distorted square-planar [Cu(**L^10^**)] units (*τ*_4_ = 0.23) [9] with Cu···Cu distance of 7.269 Å. The temperature dependence of magnetic susceptibilities *χ*_A_ and magnetic moments *µ*_A_ are represented in Figure 8, where *χ*_A_ and *µ*_A_ are per Cu unit. The *µ*_A_ value of 1.83 BM at 300 K is a little bit higher than the spin-only value (1.73 BM) for copper(II), *S* = ½ ion. The *µ*_A_ vs. *T* plot shows a slight gradual increase with a lowering of the temperature over the range 50–300 K and a gradual decrease on lowering the temperature from 50 to 2 K, reaching, 1.70 BM at 2 K. While the increase of the *µ*_A_ value suggests F interaction between two adjacent Cu(II) centers via the **L^10^** anion ligand, the corresponding decrease in *µ*_A_ values indicates an AF intermolecular interaction. The magnetic data were analyzed by the molecular field approximation (Equation (3) [8]), where for this series Equation (4) for the Heisenberg model for ferromagnetically coupled *S* = ½ ions (derived by Baker et al. [10]), considering the magnetic interaction between the neighboring chain molecules as *zJ’* (z = number of interacting neighbors), was used: *χ*_A_’ = *χ*_A_/{1−(2*zJ*’/*Ng*^2^*µ*_B_^2^)*χ*_A_}(3)
*χ*_A_ = (*Ng*^2^*µ*_B_^2^/4*kT*)[(1.0 + 5.7979916*x* + 16.902653*x*^2^ + 29.376885*x*^3^ + 29.832959*x*^4^ + 14.036918x^5^)/(1.0 + 2.7979916*x* + 7.0086780*x*^2^ + 8.6538644*x*^3^ + 4.5743114*x*^4^)]^2/3^ + *Nα*(4)
where *x* = *J*/2*kT*. The solid line in Figure 8 shows the calculated curve with best-fitting parameters of *g* = 2.1 (fixed), *J* = 3.5 cm^−1^, *Nα* = 60 × 10^−6^ cm^3^ mol^−1^ (fixed), and *zJ*’ = −2.9 cm^−1^.

## 3. Conclusions

In this work eight dinuclear doubly-bridged-phenoxido copper(II) complexes revealed antiferromagnetic coupling (AF) varied from strong AF interaction (−*J* = 289–145 cm^−1^) for complexes **1**–**6** to very weak AF interaction (−*J* = 3.6 & 4.6 cm^−1^) in **7** and **8**. The geometry around the central Cu(II) centers exhibits distorted square-pyramidal arrangement in the first four complexes, mixed square pyramidal and distorted square planar in **5** and **6**, and distorted pseudo octahedral (5 + 1) arrangements in **7** and **8**. Attempts were made to correlate the exchange coupling (*J*) to the structural parameters of the non-planar Cu_2_O_2_ four-membered rings. These parameters include the Cu···Cu distances (2.941–3.130 Å), Cu-O-Cu bond angles (96.93–103.44°) [69,70], Cu-O-Cu-O dihedral angles (0–29.8°), and the deviation angles of the phenoxido groups from the phenyl out-of-plane angles (11.3–71.0°) as well as the phenyl ring torsion angles Cu-O-C-C (2.7–31.3). Unfortunately, no satisfied correlations were observed between *J* and the mentioned geometrical parameters [69,70]. This is most likely due to the influence of the steric environment incorporated by the substituents into phenolate groups, and the terminal coordinated dialkylamine, and hence the geometrical arrangement around the central Cu(II) ions as well as the coplanarity of the phenolate groups with the bridging Cu_2_O_2_ moiety [53,70,71]. These factors may play a key role in reducing the efficient overlap between the *3d* Cu^2+^ and *2p* O-phenoxido magnetic orbitals bearing the unpaired electrons. In general and focusing the discussion only on Cu(II) complexes containing mono- and doubly-bridged phenoxido compounds with no other bridging ligands, our results agree with the magnetic properties determined in many phenoxido-bridged copper complexes, where relatively strong AF couplings were reported [1,54,69,70,72], regardless of the fact that some of these compounds were ferromagnetically coupled [1,54,72,73]. 

In contrast and unlike the bridged bis(phenoxido) dinuclear copper(II) complexes, where no correlation was observed between *J* and Cu–O–Cu bond angle, linear relationships were obtained in the corresponding bis(hydroxido) [74] and bis(alkoxido) [75] bridged copper(II) complexes. In addition, a similar linear correlation was reported in bis(phenoxido) macrocyclic complexes in which the unique conjugated π-electron inherited into the macrocycle skeleton constrained the ligand to adopt a planar configuration [71]. On the other hand, a relatively moderate AF coupling (-*J* = 59 cm^−1^) was evaluated in the trinuclear **9**, where alternative bridged phenoxido and methoxido groups were determined. However, in the 1D polymer **10**, a weak ferromagnetic interaction (*J* = + 3.5 cm^−1^) was obtained. Interestingly, in the latter complex, the Cu(II) centers were linked via the *N*,*N*-dimethylpropyl arms with long Cu···Cu distance (7.269 Å) and a tetrahedral distorted square planar four-coordinate geometry around each Cu(II) center [30].

## 4. Experimental

### 4.1. Materials and Physical Measurements

*N*,*N*-Dimethyl-, *N*,*N*-diethyl-, *N*,*N*-isopropyl-ethylenediamine, 3-dimethylamino-propylamine, 2,4-dimethylphenol, 4-chloro-2-methylphenol, 2-bromo-4-methylphenol, 2-methoxy-4-methylphenol, 4-chloro-2-methoxyphenol and 2-*tert*-butyl-4-methylphenol, picolylamine as well as dipicolylamine were purchased from TCI-America. All other chemicals were commercially available and used without further purification. 

Electronic spectra were recorded using an Agilent 8453 HP diode array UV-Vis spectrophotometer. Infrared spectra were recorded on a Cary 630 (ATR-IR) spectrometer. ^1^H spectra were obtained at room temperature on a Varian 400 NMR spectrometer operating at 400 MHz (^1^H). ^1^H NMR chemical shifts (δ) are reported in ppm and were referenced internally to residual solvent resonances (*d_6_*-DMSO: δ_H_ = 2.49) or TMS. ESI-MS of organic compounds and their Cu(II) were measured in MeOH and CH_3_CN, respectively, on a LC-MS Varian Saturn 2200 Spectrometer. Conductivity measurements were performed using a Mettler Toledo Seven Easy conductivity meter and calibrated by 1413 μS/cm conductivity standard. The molar conductivity of the complexes was determined from Λ_M_ = (1.0 × 10^3^ κ)/[Cu(II)], where κ = specific conductance and [Cu^II^] is the molar concentration of the complex. Elemental microanalyses were carried out by Atlantic Microlaboratory, Norcross, Georgia U.S.A.

The temperature dependent magnetic susceptibilities were measured over 2–300 K at the constant field of 0.5 T with a Quantum Design MPMS 3 (installed at Shimane University) for **1**–**3**, **4**, **6**–**10**, and with a Quantum Design MPMS-7 (installed at Institute for Molecular Science (IMS)) for **5**. The measured data were corrected for diamagnetic contributions [76].

### 4.2. Synthesis of the Organic Ligands

The new ligands **H_2_L^3^**, **H_2_L^4^** and **H_2_L^9^** illustrated in Figure 1, and their characterization are given below, whereas the rest of the ligands shown in this scheme (**H_2_L^1^**, **H_2_L^2^**, **H_2_L^5^**, **H_2_L^6^**, and **H_2_L^10^** as well as **HL^7^** and **H_2_L^8^**) were recently reported [30].

#### 4.2.1. 6,6′-(((2-(Dimethylamino)ethyl)azanediyl)bis(methylene))bis(2-bromo-4-methylphenol) (**H_2_L^3^**)

To a mixture containing 2-bromo-4-methylphenol (3.740 g, 20 mmol), Et_3_N (2.04 g, 20 mmol) and aqueous 37% HCHO (1.63 g, 20 mmol) dissolved in methanol (50 mL), *N*,*N*-dimethylethylenediamine (0.882 g, 10 mmol). The mixture was stirred and refluxed gently for 3 days. Evaporating the resulting solution under reduced pressure resulted in the formation of white crystalline compound, which was then filtered, washed with Et_2_O, and air dried (yield: 4.35 g, 89.5%). Characterization: calcd for C_20_H_26_Br_2_N_2_O_2_ (MM = 486.241 g/mol): C, 49.40; H, 5.39; N, 5.76%. Found: C, 49.44; H, 5.51; N, 5.78% m.p. 189° C. IR bands (ATR, cm^−1^): 2972 (w), 2950 (vw), 2820 (w), ν(C-H); 1480 (s), 1458 (vs), 1446 (s), 1374 (s), 1291 (s), 1231 (s), 1204 (s) ν(C=C, C-N, C-O); 1165 (s), 1122 (s), 1047 (m), 924 (s), 902 (s), 809 (vs). ESI-MS: *m*/*z* = 487.042, calcd [**H_2_L^6^** + H]^+^ = 487.042. 

#### 4.2.2. 6,6′-(((2-(Diethylamino)ethyl)azanediyl)bis(methylene))bis(2,4-dimethylphenol) (**H_2_L^4^**)

This compound was synthesized using a similar procedure and the same molar ratios as that described above for **H_2_L^3^**, except 2,4-dimethylphenol (2.44 g, 20 mmol) and *N*,*N*-dimethylethylenediamine (1.162 g, 10 mmol) were used instead of 2-bromo-4-methylphenol and *N*,*N*-dimethylethylenediamine, respectively (yield: 2.57 g, 66.8%). Characterization: m.p. 137–140° C. Anal. calcd for C_24_H_36_N_2_O_2_ (MM = 384.555 g/mol): C, 74.96; H, 9.44; N, 7.28%. Found: C, 74.81; H, 9.56; N, 7.28%. IR bands (ATR, cm^−1^): 2978 (vw), 2937 (vw), 2913 (vw), 2904 (m), 2802 (w) ν(C-H); 1484 (vs), 1376 (m), 1224 (vs) ν(C=C, C-O); 1152 (s), 1129 (m), 1096 (m), 1024 (m), 922 (m), 864 (s), 805 (m), 743 (vs). ESI-MS: *m*/*z* = 585.286 (100%), calcd [**H_2_L^2^** + H]^+^ = 585.285. ^1^H NMR (*d_6_*-DMSO, 400 MHz, δ in ppm): 6.84, 6.66 (2H, 2s, 1:1, ph); 3.55 (2H, s, CH_2_-ph); 2.59 (2H, m, N-*CH_2_*-CH_3_); 2.40 (4H, m, N-CH_2_-CH_2_-N); 2.14, 2.07 (3H, 2s, 1:1, 2xCH_3_-ph); 0.96 (3H, t, *CH_3_*-CH_2_). ^13^C NMR: 152 (HO-Cph); 131.0, 128.2, 127,3, 125.39, 121.44 C-ph); 55.8 (N-CH_2_ph); 49.8, 48.7 (N-CH_2_-CH_2_-N); 45.7 (CH_2_CH_3_); 20.4 (H_3_C4ph); 16.1 (H_3_C2ph); 9.9 (CH_2_CH_3_).

#### 4.2.3. 6,6′-(((2-(Diethylamino)ethyl)azanediyl)bis(methylene))bis(4-chloro-2-methylphenol) (**H_2_L^9^**)

A mixture of 4-chloro-2-methylphenol (2.825 g, 20 mmol), Et_3_N (2.022 g, 20 mmol), aqueous 37% HCHO (1.627 g, 20 mmol) and N,N-diethylethylenediamine (1.162 g, 10 mmol) was dissolved in methanol (60 mL). The mixture was stirred and refluxed gently for 3 days. Evaporating this solution under reduced pressure resulted in the formation of white precipitate, which was then collected by filtration, washed with Et_2_O, and air dried (yield: 2.35 g, 55.3%). Characterization: m.p. 137–141 °C. Cacd for C_22_H_30_Cl_2_N_2_O_2_ (MM = 424.168 g/mol): C, 62.12; H, 7.11; N, 6.59%. Found: C, 62.29; H, 7.21; N, 6.53%. IR bands (ATR, cm^−1^): 2981 (w), 2915 (vw), 2814 (w) ν(C-H); 1583 (w), 1467 (vs), 1378 (m), 1324 (m), 1273 (s), 1226 (vs) ν(C=C, C-O); 1126 (m), 1093 (s), 1063 (m), 1023 (m), 976 (m), 929 (m), 864 (vs), 804 (m) 739 (vs), 669 (m). ESI-MS: *m*/*z* = 425.175 (100%), calcd [**H_2_L^4^** + H]^+^ = 425.176. ^1^H NMR (*d_6_*-DMSO, 400 MHz, δ in ppm): 7.05, 7.00 (2H, 2s, 1:1, 2xHph); 3.58 (2H, s, CH_2_-ph); 3.37 (2H, m, N-*CH_2_*-CH_3_); 2.44 (3H, t, N-CH_2_-CH_2_-N); 2.11 (3H, s, CH_3_-ph); 0.97 (3H, t, C*H_3_*-CH_2_−).

### 4.3. Synthesis of Copper(II) Complexes

A general method was used to synthesize the copper(II) complexes **3**, **4**, and **9**: to a mixture containing the appropriate tripod phenolate-amine ligand (0.5 mmol) and Et_3_N (0.102 g, 1.00 mmol) dissolved in MeOH (20–30 mL), Cu(NO_3_)_2_·3H_2_O (0.122 g, 0.5 mmol) was added, and the resulting solution was heated for 5–10 min, filtered while hot through celite and allowed to crystallize at room temperature. The precipitate, which was collected by filtration, was washed with Et_2_O and dried in air. Crystals suitable for X-ray analysis were obtained from dilute methanolic solutions or by recrystallization from CH_3_CN, but acetone was used in complex **4**. 

#### 4.3.1. [Cu_2_(µ_2_-L^3^)_2_] (**3**)

The reaction of Cu(NO_3_)_2_·3H_2_O with H_2_L^3^ and Et_3_N (1:1:2) in MeOH, as indicated above, resulted in the formation a crude product, which upon recrystallization from CH_3_CN, afforded large dark green crystalline compound (yield: 81.6%). Anal: calcd for C_40_H_48_Br_4_Cu_2_N_4_O_4_ (MM = 1089.90 g/mol): C, 43.85; H, 4.42; N, 5.11%. Found: C, 44.03; H, 4.47; N, 5.42%. IR bands (ATR, cm^−1^): 2969 (vw), 2860 (vw), 2831 (vw), 2784 (vw) ν(C-H); 1603 (w), 1464 (vs), 1378 (w), 1278 (m), 1238 (s) ν(C=C, C-O, C-N); 1125 (m), 839 (s), 795 (s), 768 (s), 594 (m), 569 (m), 493 (m), 467 (m), 411 (s). ESI-MS (CH_3_CN): *m*/*z* = 487.042. (100%), calcd [**H_2_L^3^** + H]^+^ = 487.042. UV-VIS: λ_max_ in nm (ε_max_, M^−1^cm^−1^): in DMSO: 452 (2330), 763 (216, b). Λ_M_ (DMSO) = 3.0 Ω^−1^·cm^2^·mol^−1^. 

#### 4.3.2. [Cu_2_(µ_2_-L^4^)_2_]·CH_3_COCH_3_ (**4**)

This complex was isolated as big chunks of light olive-green solid (yield: 98 mg, 82.9%.). Recrystallization from acetone afforded X-ray quality crystals. Anal: calcd for C_51_H_74_Cu_2_N_4_O_5_ (MM = 948.425 g/mol): C, 64.46; H, 7.85; N, 5.90%. Found: C, 64.62; H, 7.88; N, 6.03%. IR bands (ATR, cm^−1^): 2966 (w), 2903 (w), 2840 (w) ν(C-H); 1609 (w) ν(C=N); 1479 (s), 1377 (m), 1319 (s), 1243 (s) ν(C=C, C-O); 1158 (m), 1091 (m), 1017 (m), 972 (m), 855 (vs), 794 (s), 751 (s), 729 (m). ESI-MS (CH_3_CN): 385.285 (100%), calcd [**H_2_L^4^** + H]^+^ = 385.286. UV-VIS in acetone: λ_max_ in nm (saturated solution): 409, 630 (b). 

#### 4.3.3. [Cu_3_(µ_2_-L^9^)_2_(µ-OCH_3_)_2_]·CH_3_OH (**9**)

The complex was isolated as shiny brownish-green crystalline compound atter recrystallization from acetone (yield: 69.3%). Anal: calcd for C_47_H_66_Cl_4_Cu_3_N_4_O_6_ (MM = 1131.500 g/mol): C, 49.89; H, 4.88; N, 4.95%. Found: C, 50.44; H, 5.16; N, 5.37%. IR bands (ATR, cm^−1^): 2976 (w), 2922 (vw), 2845 (w) ν(C-H); 1585 (w) ν(C=N); 1462 (vs), 1285 (s), 1243 (vs) ν(C=C, C-O, C-N); 1010 (m), 935 (w), 864 (s), 766 (vs), 747 (s), 661 (m), 449 (s). ESI-MS: *m*/*z* = 425.176 (100%), calcd [**H_2_L^9^** + H]^+^ = 425.176. UV-VIS: λ_max_ in nm (ε_max_, M^−1^cm^−1^): in DMSO: 447 (2650), 748 (251, b). Λ_M_ (DMSO) = 3.5 Ω^−1^·cm^2^·mol^−1^.

### 4.4. X-ray Crystal Structure Analysis

The X-ray single-crystal data of the title compounds **3**, **4**, and **9** were collected on a Bruker-AXS APEX II CCD diffractometer at 100 (2) K. The crystallographic data, conditions retained for the intensity data collection and some features of the structure refinements are listed in Table 3. Data collections were performed with Mo-K*α* radiation (λ= 0.71073 Å); data processing, Lorentz-polarization, and absorption corrections were performed using APEX and the SADABS computer programs [77,78,79]. The structures were solved by direct methods and refined by full-matrix least-squares methods on F^2^, using the SHELX [80,81,82] program library. Additional programs used: Mercury and PLATON [83]. Packing plots are given in the Appendix A).

## Data Availability

Data is contained within the available article and Appendix A.

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
