# Peer review of "Magnetostructural Properties of Some Doubly-Bridged Phenoxido Copper(II) Complexes"

_molecules, 2023, doi:10.3390/molecules28062648_

Round 1

Reviewer 1 Report

I recommend the authors to rewrite their conclusion. If it takes some time, please withdraw the manuscript and resubmit it.

1. I feel section 3 is most problematic. The authors try to correlate the observed J values to only the two parameters, Cu ... Cu distances and Cu-O-Cu bond angles. Such correlations were proposed in the very old paper (McGregor et al., Inorg. Nucl. Chem. Lett. 9, 423-428 (1973). ), where a few complexes with the planar Cu2O2 unit were compared without determining the proton positions of the OH groups. As explained in ref. [70] by the DFT calculations, there were more critical geometric parameters that affect the J value. Since the ligand orbitals of PhO group are directional, the Cu-O-Cu-O dihedral angle, the phenyl out-of plane angle, and the phenyl ring torsion angle Cu-O-C-C changes the J value. The authors would better calculate the parameters in Table 1 of ref. [70] for their complexes.

2. The Cu ... Cu distance may only be important when there are direct metal d-d interactions. According to Pantazis, J. Chem. Theory Comput. 15, 938-948 (2019), the direct metal-metal interaction is possible only for the exceptional case, like a tris-hydroxy-bridged Cr(III)-Cr(III) system, where the Cr ... Cr distance is short (0.27nm). If you consider such interactions, some evidences are needed.

3. The authors employed the parameter t  derived from the two X1-Cu-X2 angles, where X1 and X2 are ligand atoms. However, to determine the orientations of magnetic dx2-y2 orbitals, the information on the bond distances are needed. Ref. [30] shows that in the complexes 7 and 8 (1 and 2 in ref. [30]), one of the Cu-OPh bonds are0.22nm, much longer than the other Cu-OPh bond and the three Cu-N bonds. Is the dx2-y2 orbital perpendicular to the longer Cu-OPh bond? What do you expect the J value in that case?  In complexes 5 and 6 , the two Cu have different coordination numbers, so that the two magnetic d orbitals have different energies. What do you expect the J value for these cases? Prof. Franz A. Mautner may know these, so  please discuss these with him. Some more additional computational results are welcome.

4. I found that the abstract is very difficult to read. Six ligands have been already synthesized in ref. [30] and the X-ray structures of the complexes were reported there. Only three ligands are newly synthesized in this paper. All the magnetic measurements were newly made. These should be mentioned clearly. You would better find the notion other than 1-9, with which we can imagine the complexes studied easily. 

  5. There are some misspellings. The checks by coauthors are needed.

Line 38 : exgeneous --> exgenous

Line 519 : 4.3.2. --> 4.3.3.

Line 752 : Safaei, I. --> Safaei, E.

Line 756 : Hatfiel --> Hatfield

Author Response

Reviewer #1: 

I recommend the authors to rewrite their conclusion. If it takes some time, please withdraw the manuscript and resubmit it.

The corresponding author, SSM is very grateful to this reviewer with his great effort in revising Ms an the very positive comments/suggestion addressed.  This is an excellent example for a professional reviewer.  Thank you.

1.    I feel section 3 is most problematic. The authors try to correlate the observed J values to only the two parameters, Cu ... Cu distances and Cu-O-Cu bond angles. Such correlations were proposed in the very old paper (McGregor et al., Inorg. Nucl. Chem. Lett. 9, 423-428 (1973). ), where a few complexes with the planar Cu2O2 unit were compared without determining the proton positions of the OH groups. As explained in ref. [70] by the DFT calculations, there were more critical geometric parameters that affect the J value. Since the ligand orbitals of PhO group are directional, the Cu-O-Cu-O dihedral angle, the phenyl out-of plane angle, and the phenyl ring torsion angle Cu-O-C-C changes the J value. The authors would better calculate the parameters in Table 1 of ref. [70] for their complexes.

Response: The reviewer is absolutely right. The parameters Cu-O-Cu-O dihedral angle, the phenyl out-of plane angle, and the phenyl ring torsion angle Cu-O-C-C were calculated and inserted in Table 2 but unfortunately did not provide a satisfied correlation with the J values. The ligand orbitals of PhO group was also addressed and discussed in the magnetic section 3.

2.    The Cu ... Cu distance may only be important when there are direct metal d-d interactions. According to Pantazis, J. Chem. Theory Comput. 15, 938-948 (2019), the direct metal-metal interaction is possible only for the exceptional case, like a tris-hydroxy-bridged Cr(III)-Cr(III) system, where the Cr ... Cr distance is short (0.27nm). If you consider such interactions, some evidences are needed.

Response: We agree with the reviewer with this point and was addressed in the last paragraph of the conclusion section in the first version of the Ms.

3.    The authors employed the parameter t  derived from the two X1-Cu-X2 angles, where X1 and X2 are ligand atoms. However, to determine the orientations of magnetic dx2-y2 orbitals, the information on the bond distances are needed. Ref. [30] shows that in the complexes 7 and 8 (1 and 2 in ref. [30]), one of the Cu-OPh bonds are 0.22nm, much longer than the other Cu-OPh bond and the three Cu-N bonds. Is the dx2-y2 orbital perpendicular to the longer Cu-OPh bond? What do you expect the J value in that case?  In complexes 5 and 6 , the two Cu have different coordination numbers, so that the two magnetic d orbitals have different energies. What do you expect the J value for these cases? Prof. Franz A. Mautner may know these, so  please discuss these with him. Some more additional computational results are welcome.

Response:  This is a good point, and we are very thankful to this reviewer for his true and sincere effort in revising the Ms. The differences in Cu-O-Cu were addressed and discussed in all compounds. In addition, the orientations of magnetic dx2-y2 orbitals were carefully addressed in each group of compounds.

4.    I found that the abstract is very difficult to read. Six ligands have been already synthesized in ref. [30] and the X-ray structures of the complexes were reported there. Only three ligands are newly synthesized in this paper. All the magnetic measurements were newly made. These should be mentioned clearly. You would better find the notion other than 1-9, with which we can imagine the complexes studied easily.

Response: We also, agree.  We changed the notion of the compounds and made some changes in the abstract showing clearly that we have three new ligands and their corresponding complexes, which were synthesized in this work, but the rest have been already synthesized before as indicated in Ref [30].  Thank you.

5.    There are some misspellings. The checks by coauthors are needed
Line 38 : exgeneous --> exgenous
Line 519 : 4.3.2. --> 4.3.3.
Line 752 : Safaei, I. --> Safaei, E.
Line 756 : Hatfiel --> Hatfield

Response: These were fixed. Thank you and our apology for these misspellings.

Reviewer 2 Report

Reviewer comments

This work reports synthesis and characterization of a series of copper(II) complexes containing  tripod tetradentate phenolate-amine ligands. The 26 temperature dependence magnetic susceptibility study over the temperature range 2-300 K was performed. Such studies in the year 2023 are traditional, especially since the work does not include any applied value for the copper(II) complexes under study. However, the manuscript needs careful review, especially the language and wording, and I ask the authors to take the following comments into consideration during the preparation and re-submission of the manuscript for review.

- Page 1, line 37, the word cooper should change to copper and please keep that correction in mind throughout the manuscript.

- Page 12, line 39; what does (5+1) mean?

- The wording copper(II) atoms should change to copper(II) centers or ions and please keep that correction in mind throughout the manuscript.

- The chromophores Cu2O2 and N2O2 should change to Cu2O2 and N2O2 and please keep that correction in mind throughout the manuscript.

- The magnetic moment unites is Bohr magneton (BM) and the abbreviation of magnetic moment is μM, μs, μl or μeff. But what is written in the manuscript is strange to me as a reviewer. I request the authors to take this into account and make the necessary correction throughout the manuscript.

- The section discuses description of the structures of complexes on pages 5 and 6 needs revision and rewriting. This is because there is ambiguity, broadcast and non-arrangement of wording in describing the structure of the current copper(II) complexes which makes it difficult to understand what the authors want to present.

- For the square pyramidal geometry, equatorial (basal) plane donor sites must be specified for all structures.

- The crystal structure of title compounds 3 and 4.CH3COCH3 consists of neutral dinuclear [Cu2(L3)2] respectively [Cu2(L4)2] units; the latter co-crystallizes with an acetone solvent molecule per dinuclear unit (Figure 1a and 1b). For this sentence, it is not clear to me the reason for the existence of the syllable CH3COCH3. I think it is sufficient to mention the number of the metal complex and there is no need for that so as not to cause confusion to the reader and this is taken into account throughout the manuscript for similar cases.

- The section (2.5.), which discusses magnetic properties, is very long and repetitive, and reduction is preferred.

In light of the foregoing, as a reviewer, I cannot recommend the acceptance of this manuscript in its present state for publication in the Journal molecules.

Author Response

Reviewer #2: 

This work reports synthesis and characterization of a series of copper(II) complexes containing  tripod tetradentate phenolate-amine ligands. The 26 temperature dependence magnetic susceptibility study over the temperature range 2-300 K was performed. Such studies in the year 2023 are traditional, especially since the work does not include any applied value for the copper(II) complexes under study. However, the manuscript needs careful review, especially the language and wording, and I ask the authors to take the following comments into consideration during the preparation and re-submission of the manuscript for review.
Yes, we do agree about the addressed point but our intention was focused on finding a correlation between large group of doubly bridged-diphenoxido compounds with the J values but this was not the case, properly the substituents incorporated at two sides of the ligand were not working in a harmonic way.  It seems, that the planarity of the bridging phenoxido groups within the Cu2O2 ring is very crucial for observing such correlation.

- Page 1, line 37, the word cooper should change to copper and please keep that correction in mind throughout the manuscript.

Response: This was fixed. Our apology, thank you.

- Page 12, line 39; what does (5+1) mean?

Response: The 5+1 notation is used in X-ray crystallography to indicate pseudo six-coordinate species, specially when five coordinating centers are clearly coordinated and the sixth coordination site has a longer bond, as in our compounds 7 and 8 which show semi coordination of the O-Me group.

- The wording copper(II) atoms should change to copper(II) centers or ions and please keep that correction in mind throughout the manuscript.

Response:  Thank you.  These were changed throughout the Ms

- The chromophores Cu2O2 and N2O2 should change to Cu2O2 and N2O2 and please keep that correction in mind throughout the manuscript.

Response: These were fixed.  Thank you, 

- The magnetic moment unites is Bohr magneton (BM) and the abbreviation of magnetic moment is μM, μs, μl or μeff. But what is written in the manuscript is strange to me as a reviewer. I request the authors to take this into account and make the necessary correction throughout the manuscript.

Response: The reviewer is right.  These were fixed.  Thank you for paying our attention to this point.

- The section discuses description of the structures of complexes on pages 5 and 6 needs revision and rewriting. This is because there is ambiguity, broadcast and non-arrangement of wording in describing the structure of the current copper(II) complexes which makes it difficult to understand what the authors want to present.

Response: This section was revised as the author indicated. 

- For the square pyramidal geometry, equatorial (basal) plane donor sites must be specified for all structures.

Response: These were defined and fixed.

- The crystal structure of title compounds 3 and 4.CH3COCH3 consists of neutral dinuclear [Cu2(L3)2] respectively [Cu2(L4)2] units; the latter co-crystallizes with an acetone solvent molecule per dinuclear unit (Figure 1a and 1b). For this sentence, it is not clear to me the reason for the existence of the syllable CH3COCH3. I think it is sufficient to mention the number of the metal complex and there is no need for that so as not to cause confusion to the reader and this is taken into account throughout the manuscript for similar cases.

Response: we were hesitating to make these numbering without solvents, but changes were made and it is fine with us. 

- The section (2.5.), which discusses magnetic properties, is very long and repetitive, and reduction is preferred.

Response: Yes, the reviewer is right.  This section was reduced as much as we can but because inserting some new data, suggested by reviewers, it became long again.  One of the points which made it long is the fact since the molecular structures six complexes were not described, then when the magnetic properties were divided into groups, we did not have choices other than providing a brief description for the crystal structures of each group in order to account for their magnetic behaviors. 

In light of the foregoing, as a reviewer, I cannot recommend the acceptance of this manuscript in its present state for publication in the Journal molecules.

Response:  Thank you for your effort and your opinion is fully respected and hope the revised version is better than before,

Reviewer 3 Report

The following article entitled ‘Magnetostructural Properties of Some Doubly Bridged

Phenoxido Copper(II) Complexes’ authored by Massoud et al demonstrates the synthesis of a series of Cu(II) complexes followed by their crystallographic and magnetic property study. To me this work does not appear to be quite impressive, instead it looks like a routine process without any new insight. Below are my comments:

  1. The manuscript needs extensive English language editing. In many places the sentences appear to be incomplete and are not easily understandable what the authors want to say.

  2. Abstract and conclusion sections look repetitive, these can be modified. In line 25 and line 390 the four coordinate geometry (the key word) looks missing. In line 41 hemocyanin is said as metalloenzyme, I think this metalloprotein is not an enzyme.

  3. Scheme 1 is the pictorial representation of the ligands but some where it was written as synthetic procedure which can be modified.

  4. In the abstract the authors report that ‘No correlation was found between the J values with the bond angles (96.93 – 103.44°) nor the Cu···Cu distances (2.941 – 3.130 Å).’ and in line 59 it is written as ‘The major factors affecting the exchange interactions are the Cu‧‧‧Cu bond distance and the bridged Cu-O-Cu bond angle’. These two lines can be confusing, it needs clarification or it should be rephrased.

  5. I feel that the introduction should be more rich with more background and information about why this area of study is important. Overall the manuscript lacks the key thing about why people will be interested in this research or what is the takeaway of this. This information should be there in the conclusion section.

Author Response

Reviewer #3: 

The following article entitled ‘Magnetostructural Properties of Some Doubly Bridged
Phenoxido Copper(II) Complexes’ authored by Massoud et al demonstrates the synthesis of a series of Cu(II) complexes followed by their crystallographic and magnetic property study. To me this work does not appear to be quite impressive, instead it looks like a routine process without any new insight. Below are my comments:

Response: Thank you, we fully respect your opinion. In fact, our intention was focused on finding a correlation between large group of doubly bridged-diphenoxido compounds with the J values but this was not the case, properly the substituents incorporated at two sides of the ligand were not working in a harmonic way.  It seems, that the planarity of the bridging phenoxido groups within the Cu2O2 ring is very crucial for observing such correlation.

1.    The manuscript needs extensive English language editing. In many places the sentences appear to be incomplete and are not easily understandable what the authors want to say.

Response: The English was revised in this version.

2.    Abstract and conclusion sections look repetitive, these can be modified. In line 25 and line 390 the four coordinate geometry (the key word) looks missing. In line 41 hemocyanin is said as metalloenzyme, I think this metalloprotein is not an enzyme.

Response: We agree with all addressed points.  These were fixed.

3.    Scheme 1 is the pictorial representation of the ligands but some where it was written as synthetic procedure which can be modified.
Response: We did not understand this point.  Therefore, we left the scheme as it is. 

4.    In the abstract the authors report that ‘No correlation was found between the J values with the bond angles (96.93 – 103.44°) nor the Cu···Cu distances (2.941 – 3.130 Å).’ and in line 59 it is written as ‘The major factors affecting the exchange interactions are the Cu‧‧‧Cu bond distance and the bridged Cu-O-Cu bond angle’. These two lines can be confusing, it needs clarification or it should be rephrased.

Response: The abstract and conclusions were revised and modified due to the incorporation of other parameters, which may affect the magnetic properties.

5.    I feel that the introduction should be more rich with more background and information about why this area of study is important. Overall the manuscript lacks the key thing about why people will be interested in this research or what is the takeaway of this. This information should be there in the conclusion section.

Response: Our apology, we do not agree with the reviewer in this point. The Ms covered the necessary background related to the addressed compounds. However, some changes were made in the introduction and conclusions.

Round 2

Reviewer 1 Report

The paper has been much improved from the former  version. I agree that it will be published in this form.

Author Response

Reviewer 1

The paper has been much improved from the former  version. I agree that it will be published in this form.

Thank you,

Reviewer 2 Report

The corrections made by the authors are acceptable, but there are still some that need correcting as follows:

- Page 1, line 37, the word cooper should change to copper

- Page 5, line, formula Cu1‧‧‧Cu2 should change to Cu(1)…Cu(2), as well, N1 and N2 should change to N(1) and N(2), … and so on. Please consider this in all similar cases throughout the manuscript as this is usual in the case of describing structural analysis of metallic complexes by X-rays.

- Page 12, line 378, [Cu(L-10)] should change to [Cu(L10)]

- English needs a final revision to adjust some sentences and meanings.

In the event that the authors do the required work, the manuscript can be accepted for publication in the Journal molecules.

Author Response

Reviewer 2

The corrections made by the authors are acceptable, but there are still some that need correcting as follows:

We thank the reviewer again.

- Page 1, line 37, the word cooper should change to copper

Thank you.  This was fixed.

- Page 5, line, formula Cu1‧‧‧Cu2 should change to Cu(1)…Cu(2), as well, N1 and N2 should change to N(1) and N(2), … and so on. Please consider this in all similar cases throughout the manuscript as this is usual in the case of describing structural analysis of metallic complexes by X-rays.

In fact, this is a very minor point, there is no standard way of how write Cu   Cu  or any related bond distance.  We have published more than 140 papers describing X-ray structures of many metal complexes and no one raised this issue and the reviewer indicated that “this is usual in the case of describing structural analysis of metallic complexes by X-rays”.  We think this is his/her  opinion. Therefore, we prefer to use the formula  Cu1‧‧‧Cu2 without changing it to Cu(1)…Cu(2).  However, we fully respect his opinion.

- Page 12, line 378, [Cu(L-10)] should change to [Cu(L10)]

Thank you.  This was fixed.

- English needs a final revision to adjust some sentences and meanings.

In fact, this is a very general statement, and no indication was addressed to any specific point, otherwise reviewers 1 and 3 would refer to English revision.

In the event that the authors do the required work, the manuscript can be accepted for publication in the Journal molecules.

Reviewer 3 Report

The revised manuscript authored by Massoud et. al. entitled “Magnetostructural Properties of Some Doubly Bridged Phenoxido Copper(II) Complexes” is significantly improved from the initial submission. The authors have modified the manuscript as recommended and all the comments have been taken care of where they are agreed with. In my opinion it can be published here in its current form.

Author Response

Reviewer 3

The revised manuscript authored by Massoud et. al. entitled “Magnetostructural Properties of Some Doubly Bridged Phenoxido Copper(II) Complexes” is significantly improved from the initial submission. The authors have modified the manuscript as recommended and all the comments have been taken care of where they are agreed with. In my opinion it can be published here in its current form

Thank you
